# Multidrug-Resistant Micro-Organisms Associated with Urinary Tract Infections in Orthopedic Patients: A Retrospective Laboratory-Based Study

**DOI:** 10.3390/antibiotics10010007

**Published:** 2020-12-23

**Authors:** Grzegorz Ziółkowski, Iwona Pawłowska, Michał Stasiowski, Estera Jachowicz, Jadwiga Wójkowska-Mach, Tomasz Bielecki

**Affiliations:** 1Sosnowiec Medical College, Wojska Polskiego 6 Str., 41-200 Sosnowiec, Poland; nc3@wp.pl; 2Division of Microbiology and Epidemiology, St. Barbara Specialised Regional Hospital No. 5, Medyków 1 Square, 41-200 Sosnowiec, Poland; ivi5@op.pl; 3Clinical Department of Anaesthesiology and Intensive Therapy, Faculty of Medical Sciences in Zabrze, Medical University of Silesia, 40-055 Katowice, Poland; mstasiowski.anest@gmail.com; 4Department of Anaesthesiology and Intensive Therapy, St. Barbara’s Memorial Regional Hospital in Sosnowiec, Plac Medyków 1, 41-200 Sosnowiec, Poland; 5Department of Microbiology, Faculty of Medicine Jagiellonian University Medical College, 31-121 Kraków (Cracow), Poland; jadwiga.wojkowska-mach@uj.edu.pl; 6Department of Orthopedics of the Faculty of Medicine with the Division of Dentistry in Zabrze, Medical University of Silesia, 40-055 Katowice, Poland; tbiel@tlen.pl

**Keywords:** urinary tract infections, orthopedics, surveillance, multi-drug-resistant micro-organisms

## Abstract

**Background:** The risk of healthcare-associated infections (HAIs) in surgical wards remains closely related to the type of surgery and procedures performed on patients. Those factors also condition the risk of various forms of clinical infections, especially urinary tract infections (UTIs). UTIs are most frequently (70–80% of cases) caused by the use of bladder catheter in the perioperative period. The aim of this study was to perform an epidemiological and microbiological analysis of UTIs in orthopedic patients, with an emphasis on multidrug-resistant (MDR) micro-organisms. **Methods:** The study was conducted in a 38-bed Department of Orthopedic-Traumatic Surgery in Sosnowiec, Poland. 5239 patients, operated on in 2013–2015, were included in the study. The urinary catheter use rate was 30.7%. Laboratory-based study used the UTI definition of the HAI-Net program. A micro-organism was declared MDR if it was resistant to at least one antibiotic from three or more groups of antibacterial drugs, and extensively drug-resistant (XDR) if it was sensitive to antibiotics from no more than two groups of drugs. **Results:** The UTI incidence was 3.2% (168 cases), the CA-UTI incidence density was 9.6/1000 catheter days. The highest risk of UTI was found in patients aged 75 or older. Monomicrobial cultures were detected in 163 specimens (78% of all microbiologically confirmed UTIs). Gram-negative flora prevailed among the micro-organisms, the predominantly isolated *Enterobacteriaceae* being *Escherichia*
*coli* and *Klebsiella*
*pneumoniae*. In 16 patients (7.7% of microbiologically confirmed UTIs), yeast infection was confirmed. Isolated micro-organisms were fully sensitive to carbapenems. Gram-negative bacilli showed the lowest sensitivity to extended substrate spectrum penicillins and fluoroquinolones (37–64%), as well as to trimethoprim-sulfamethoxazole (50%). The MDR prevalence was 24.4%. **Conclusions:** The presented data indicates that UTIs are a significant problem in the studied population, so is antimicrobial resistance, especially to quinolones, and extended-spectrum cephalosporins, which are often used as first-line therapy. To tackle the problem of high UTI incidence and MDR prevalence, reducing the UTI risk factors should be prioritized.

## 1. Background

The risk of healthcare-associated infections (HAIs) in surgical wards remains closely related to the type of surgery and procedures performed on patients. Those factors also condition the risk of various forms of clinical infections, especially the urinary tract infections (UTIs). According to Gossett et al., patients who developed a UTI after total hip arthroplasty were more likely to be female and have more comorbidities [1]. The UTIs are most frequently (70–80% of cases) caused by the use of bladder catheter (CA-UTI, catheter-associated urinary tract infections) in the perioperative period [2]. The bladder catheterization procedure bears a risk of introducing some micro-organisms of the microbiota of urethral mucosa into the bladder. Therefore, it is important that the skin and mucous membranes be properly decontaminated prior to bladder catheterization. [3,4,5].

UTIs and CA-UTIs are caused by both gram-negative and gram-positive bacteria, as well as by some fungi, viruses, and protozoa. The gram-negative bacilli, including *Escherichia coli* and multi-drug-resistant strains, such as *Proteus mirabilis, Klebsiella pneumoniae, Pseudomonas aeruginosa, Enterobacter* spp., are responsible for about 80% of all UTI cases. Other important micro-organisms are enterococci and staphylococci [4,6]. 

UTI surveillance, control and prevention, including at an orthopedic ward, should be based on constant monitoring of UTIs, analysis of their etiological factors, including antibiotic sensitivity analysis, as well as determining the risk factors for UTI development [5].

The aim of this study was to conduct an epidemiological and microbiological analysis of UTIs in patients undergoing surgery at the studied orthopedic department, and to determine the drug sensitivity of the etiological factors of those UTIs. The authors’ experiences have already been partly discussed; however, they only concerned surgical site infections in general, without taking different infection types into account [7].

## 2. Materials and Methods

The study of UTI incidence was carried out between 1 January 2013 and 31 December 2015, at a 38-bed Clinical Department of Orthopedic, Trauma, Oncological, and Reconstructive Surgery of St. Barbara Regional Specialist Hospital No 5 in Sosnowiec, one of the largest in southern Poland. 

The urinary catheter use rate was 30.7%. Catheterization procedures were implemented both perioperatively and for other reasons. Each of the operated patients had a catheter in the perioperative period. Both insertion and removal took place within the ward, outside the operating theatre. The hospital did not use a closed bladder drainage system during the studied period. Catheter was most frequently implemented for 48 h in the perioperative period. 

The UTI cases were analyzed retrospectively, in a laboratory-based study conducted in cooperation with the Hospital Infections Control Team on the basis of microbiological surveillance, using the Healthcare-Associated Infections Surveillance Network (HAI-Net) definitions [8]. HAI-Net is a European network for HAI surveillance. The network is coordinated by the European Centre for Disease Prevention and Control (ECDC), an EU agency established in 2005 tasked with strengthening Europe’s defenses against infectious diseases. Participation in HAI-Net is voluntary and confidential for the European hospitals. The decision on urine sampling and culture was made each time by the treating physician and based on clinical symptoms. Symptomatic UTIs—without the asymptomatic bacteriuria—were diagnosed and classified as symptomatic infections (with or without microbiological confirmation) in accordance with the uniform definitions issued by the ECDC. A CA-UTI was defined as a symptomatic UTI in which the positive culture was taken after an indwelling urinary catheter had been in place for at least 48 h [8].

For the microbiological diagnosing of CA-UTI, urine samples were collected through newly placed catheters using aseptic technique. In the case of patients without catheters, mid-stream urine was collected.

Only the first isolate from each patient was selected for microbiological analysis, excluding subsequent cultures from the same patient and UTI case. The minimum inhibitory concentrations (MICs) of antimicrobials were determined by means of the broth microdilution method using the Phoenix 100 automated system (Becton Dickinson, Warsaw, Poland), and using the agar dilution method for ESBL (extended-spectrum beta-lactamases) activity identification. Combo panels NMIC/ID and PMIC/ID were used to identify and determine the antibiotic sensitivity (excluding ESBL activity). 

Antibiotic susceptibility tests were interpreted in accordance with the EUCAST recommendations for the relevant calendar year, i.e., versions 3.1 (2013), 4.0 (2014) and 5.0 (2015) [9]. Intermediately sensitive isolates for a given antibiotic were classified as resistant [10]. ESBL activity was detected with a modified double disk synergy test using a combination of ceftriaxone (30 μg, BD), cefotaxime (30 μg, BD), ceftazidime (30 μg, BD), aztreonam (30 μg, BD), and amoxicillin/clavulanic acid (20/10 μg, BD), in compliance with the EUCAST recommendations. The four antibiotics were placed at the distances of 20 mm edge to edge from the amoxicillin/clavulanic acid disk that was placed in the middle of the plate. If, after 24 h of incubation, an enhanced zone of inhibition between either of the cephalosporin antibiotics and the amoxicillin/clavulanic acid disk occurred, the test was considered to be ESBL-positive [11].

Bacteria were classified as multidrug-resistant (MDR) if the strain was resistant to at least one antibiotic from at least three groups of antibacterial drugs, or as extensively drug-resistant (XDR) if the strain was antibiotic-sensitive to no more than two groups of drugs. 

The epidemiological analysis of the data took into consideration the following indicators [8]: UTI incidence rate, calculated as: (N of UTI × 100%)/N of operations; CA-UTI rate: N of CA-UTI*1000/N of urinary catheter days; and urinary catheter use rate: N of urinary catheter days/N of patient-days.

In the statistical analysis, the number and percentage of individual variants was used. The results were analyzed by means using the chi-square test (chi^2^) or Fisher’s exact test. Analyses of changes in individual MDR strains were analyzed by estimating Tau-Kendall’s rank correlation coefficients. Statistical significance was assumed at *p* < 0.05. We used IBM SPSS statistics (v. 24; IBM Corp., Armonk, NY, USA) for all statistical analyses.

The use of data was approved by the Bioethical Committee of Sosnowiec Medical College in Sosnowiec (No. PW/WSM/36/17). All data entered into the electronic database and analyzed in this study had been anonymized.

## 3. Results

A total of 5239 patients were operated on in the studied period, (51,170 patient-days, pds), 46% of them being female and 54% being male. 168 UTI cases were diagnosed, including 151 CA-UTIs (89.9% of all cases). Average age of the patients with UTI was 67 years (standard deviation, SD 20): 74 for women (SD 17) and 59 for men (SD 20), significantly lower in women (*p* < 0.0001).

Monomicrobial cultures were detected in 163 specimens (78.0% of all microbiologically confirmed UTIs), and polymicrobial in 46 specimens (22.0%), 24 (52.2%) in women and 22 (47.8%) in men (odds ratio (OR) 1.4, 95% confidence level (CI) 0.68–2.99, *p* = 0.43). In 5 UTI cases, the etiological factor was not isolated. Non-catheter-associated UTIs—17 cases—were microbiologically confirmed in 12 cases (70.6%), all these cases were monomicrobial. 

The UTI incidence rate was 3.2% (168 cases), and CA-UTI (151 cases) incidence density rate was 9.6/1000 catheter days. 

In non-catheter-associated UTIs were isolated *Enetrococcus faecalis* (2 cases), *Escherichia coli* (5 cases), *Klebsiella pneumoniae* (2 cases), *Proteus mirabilis* and *Candida albicans* (2 cases, Table 1).

In total, 197 strains of various pathogenic species considered to be etiological factors of CA-UTI have been analyzed. Gram-negative micro-organisms were dominant in the isolated micro-organisms (76.7% of the microbiologically confirmed CA-UTIs, Table 1). Among them, some of the most common strains were: *Enterobacteriaceae* from the *Enterobacterales* family ord nov. (*Escherichia coli, Klebsiella pneumoniae*), *Proteus mirabilis* from the *Morganellaceae* family, and non-fermenting rods—*Pseudomonas aeruginosa*. *Enterococcus faecalis* was dominant among the gram-positive flora. The prevalence of *Enterobacterales* family was significantly higher in the monomicrobial CA-UTI than in the polymicrobial: 65.2% vs. 36.5% (OR 6.9, 95% CI 3.41–14.33, *p* = 0.00012, Table 1)

In 16 patients (9.8% of microbiologically confirmed UTIs), yeast infection was confirmed: 2 cases in non-catheter-associated UTIs, and 14 cases in CA-UTI. 8 of them (50%) involved women, the yeast etiology was similar for both genders (Table 1). The prevalence of yeast etiology in non-catheter-associated UTIs was 16.7% and in CA-UTI: 9.3% (not statically significant, OR 2.6143, 95%CI 0.52–13.11, *p* = 0.112).

Isolated micro-organisms showed full sensitivity to carbapenems, linezolid, and glycopeptides over the study period. Gram-negative bacilli showed the lowest sensitivity to extended-spectrum penicillins and quinolones: 37–64%, and to trimethoprim-sulfamethoxazole: 50%. The lowest total sensitivity was shown by *K. pneumoniae* strains, which showed high resistance, in addition to the above-mentioned ones, also to the third-generation cephalosporins (cefotaxime and ceftazidime), penicillins in combination with beta-lactamase inhibitors, and aminoglycosides (Table 2). MDR strains were only isolated in CA-UTIs, on average on the 19th day after admission to hospital. The total prevalence of drug-resistant micro-organisms was 24.4% (25.9% in CA-UTIs) and remained at a similar level during the study period (*p* = 0.6015). The highest MDR prevalence was observed in the rarely isolated non-fermenting *Acinetobacter baumannii*, and the frequently isolated *K. pneumoniae*. It amounted to 77.8% and 65.6%, respectively (Table 3). The prevalence of MDR was not significantly related to gender (OR 0.714, 95% CI 0.38–1.36, *p* = 0.091).

## 4. Discussion

The observed incidence rates were very high, especially the CA-UTI incidence density rate and the MDR prevalence. UTIs are the second most common postoperative complication of patients treated at orthopedic wards, more common than deep venous thrombosis, pneumonia or renal failure [12,13]. The risk of major postoperative complications, such as a surgical site infection (SSI), following an orthopedic surgery is estimated between 1.4% and 20%, and sometimes it is even higher. In the studied center, the SSI incidence rate was 6.6%, which is about 6 times higher compared to the European HAI-Net [7]. 

Unfortunately, the presented data indicate that the UTI epidemiology also is a significant problem at the studied department, e.g., in the data of the American NHSN (National Healthcare Safety Network) program regarding the orthopedics and traumatology departments, the CA-UTI incidence density was more than three times lower [13]. In the ECDC infection surveillance program, the CA-UTI incidence rate was 4.1 per 1000 patient-days, it was then more than twice lower than at the studied ward—despite the fact that it concerned intensive care units [14]. The CA-UTI rate in this study is also significantly higher than in another comparable hospital in Poland [15].

What may have been the problem is the bladder catheterization procedure used at this ward, which has never been validated. Hence the conclusion about the need for systematic surveillance regarding UTI identification, and for proper infection prevention and control, especially when dealing with patients with indwelling catheters [5,16]. A good example of that is the intervention described by Takker et al. in the surveillance of patients undergoing total hip and knee replacement and hip fracture treatment, where morbidity was reduced from 2.1 to 1.1%. The intervention involved the implementation of catheter surveillance principles, including qualification for catheterization, which resulted in a reduction of the catheterization rate 55.2% to 19.8% [5]. The data presented in our study indicate a significantly higher proportion of catheterized patients, which is probably due to overusing the procedure in both intraoperative and postoperative period. 

In case of orthopedic patients, it has been shown that the proper use of perioperative antibiotic prophylaxis in accordance with local decision-making standards improves the patient’s prognosis in terms of SSIs and, additionally, also in the prevention of CA-UTIs [12]. Based on the current evidence, the urinary catheterization during total joint arthroplasty increases the risk of postoperative CA-UTIs, and it may not be routinely required for patients undergoing such procedures [2]. 

On the other hand, postoperative complications following anorectal, hernia, or orthopedic surgeries are quite common; they may include, e.g., urinary retention [17,18]. That is why it is difficult to decide when it is safe not to use a catheter and when its insertion is necessary. The risk of urinary retention increases with the patient’s age and is most probable in old males.

In addition to the catheterization procedure itself, many other factors contribute to the UTI development, including patient-dependent unmodifiable factors, with age being a particularly strong UTI predictor. This is confirmed by other authors [13,19] as well as by presented results, which indicate a significant, more than 3-fold higher risk of infection in patients aged 75 years and older. This may be due to a generally weaker immune response in older patients, but also a weaker physical condition or limited mobility. Consideration should be given to HAI surveillance, including UTI prevention and control in orthopedic departments, using the generally accepted scale for geriatric patients, such as the Bathel scale, which is designed to assess patients’ mobility. Such a stratified description of patients would facilitate the implementation of special surveillance of persons exposed to HAIs. 

Typical microbial virulence factors are important in the CA-UTI pathomechanism, especially the formation of abundant biofilm and urease, which are responsible for increasing the urine pH. These virulence factors are present in the *P. aeruginosa, K. pneumoniae, P. mirabilis, Morganella morganii*, and some *Providencia* spp. infections. Thus, the dominance of the gram-negative *Enterobacterales* bacteria [12,20,21] in this study may be unusual, but is well-founded. Those factors are also present in some strains of *Staphylococcus aureus* and coagulase-negative staphylococci [4,22].

The observed prevalence of yeast etiology was 16.7% and 9.3% (respectively in non-CA-UTIs and in CA-UTI), among adults in community-acquired UTIs, yeast are encountered in <1% of clean-voided urine specimens but account for 5–10% of positive urine culture results in hospitals, mostly in patients with bladder catheter [23]. So, our results were slightly higher than expected, but on the other hand, candiduria are common in surgical patients [24].

However, the described low drug sensitivity is also a significant problem. Resistance to quinolones and extended-spectrum cephalosporins remains a major challenge, because these antibiotics are widely used as first-line therapy in the treatment of UTIs. Particular consideration should be given to penicillins sensitive to beta-lactamases (due to the high proportion of ESBL+ strains), and to fluoroquinolones. The largest group of ESBLs are CTX-Ms are becoming more common worldwide, especially the CTX-M-15, which is often associated with the uropathogenic *E. coli* clone. In addition, plasmids, which often carry ESBL genes, also carry determinants of fluoroquinolone resistance [25]. Therefore, the use of both ciprofloxacin—a drug common in empirical therapy of UTIs and eagerly used empirically in Poland for bladder infections—and trimethoprim/sulfamethoxazole, carry the risk of therapeutic failure. High consumption of trimethoprim/sulfamethoxazole in Poland can be one of the reasons for high microbial resistance to it [26].

The subject gets complicated by the high prevalence of MDR strains, especially the non-fermenting *A. baumannii* and *K. pneumoniae*, which account for about 1/4 of all UTIs [27,28]. The studied department, unfortunately, represents a very typical ward. According to the figures for 2014 presented by Mazzariol et al. [28], in EU/EEA countries the percentage of ESBL(+) *K. pneumoniae* resistant strains in UTIs was over 70% (Romania, Greece), and ESBL(+) *E. coli* over 40% (Bulgaria). In Poland, ESBL(+) strains constituted 65% of *K. pneumoniae* strains and 11% of *E. coli*. In the countries of Northern Europe (Finland, Iceland), the prevalence of this type of resistant strains is significantly lower [26]. Unfortunately, at the studied unit, the high prevalence of MDR micro-organisms was observed not only in UTIs, but also in surgical site infections (22.6%). The high multi-drug resistance mainly concerned the gram-negative bacilli: *A. baumannii* and *K. pneumoniae* [7].

The main advantage our study is its usefulness for healthcare systems in the region, given the lack of previous research on the topic in Central and Eastern Europe, and a large and homogeneous population of orthopedic patients that we have studied. We have used standardized, repeated methods of drug resistance testing. Our retrospective study was cost-effective and carried out in short time. The limitations of our one-center, non-experimental study were a retrospective rather than prospective UTI analysis, and a limited ability to compare our results with local antibiotic sensitivity patterns in the population of orthopedic patients with UTIs, due to the lack of comparative data—non-surgical-site infections following orthopedic surgeries are poorly described, especially in our part of Europe. 

## 5. Conclusions

As in other publications Enterobacteriaceae bacilli from the *Enterobacterales* family were the most prevalent species, but unfortunately the prevalence of *Candida* spp. both in CA-UTI and in non-catheter-associated UTIs was also high. Regular reporting of unit-specific CA-UTI rates to clinical care staff should be the key element of CA-UTI surveillance. It is also necessary to implement an empirical therapy based on continuous antimicrobial resistance surveillance and an antimicrobial stewardship program. It would be appropriate to discuss the legitimacy of catheterization after certain procedures and overusing it, in both intraoperative and postoperative periods.

## Figures and Tables

**Table 1 antibiotics-10-00007-t001:** The most frequently isolated etiological factors in microbiologically confirmed urinary tract infections—including 36 polymicrobial cases—in the Clinical Department of Orthopedic-Traumatic, Oncological, and Reconstructive Surgery in 2013–2015.

Pathogen	Monomicrobial *n* [%]		Polymicrobial *n* [%]		Total *n* [%]		Depending on the Use of Bladder Catheter [%]		Total *n* [%]
	Female	Male	Female	Male	Female	Male	CA-UTI	Non-CA-UTI	
Gram-positive (16.3% of all micro-organisms)									
*Staphylococcus aureus*	0 (0.0)	2 (2.4)	0 (0.0)	0 (0.0)	0 (0.0)	2 (1.9)	2 (1.0)	0 (0.0)	2 (1.0)
*Enterococcus faecalis*	5 (6.4)	5 (5.9)	5 (6.5)	5 (22.7)	10 (9.8)	10 (9.3)	18 (9.1)	2 (16.7)	20 (9.6)
*Enterococcus faecium*	0 (0.0)	0 (0.0)	0 (0.0)	1 (4.5)	0 (0.0)	1 (0.9)	1 (0.5)	0 (0.0)	1 (0.5)
Others	11 (14.1)	0 (0.0)	0 (0.0)	0 (0.0)	11 (10.8)	0 (0.0)	11 (5.6)	0 (0.0)	11 (5.3)
Gram-negative (76.1% of all micro-organisms)									
*Escherichia coli*	13 (16.7)	25	15 (62.5)	8 (36.4)	28 (27.5)	33 (30.8)	56 (28.4)	5 (41.7)	61 (29.2)
*Klebsiella pneumoniae*	11 (14.1)	14	4 (16.7)	3 (13.6)	15 (14.7)	17 15.9)	30 (15.2)	2 (16.7)	32 (13.3)
*Proteus mirabilis*	7 (9.0)	13	0 (0.0)	5 (22.7)	7 (6.9)	18 (16.8)	24 (12.2)	1 (8.3)	25 (12.0)
*Pseudomonas aeruginosa*	10 (12.8)	11	0 (0.0)	0 (0.0)	10 (99.8)	11 (10.3)	21 (10.7)	0 (0.0)	21 (10.0)
*Acinetobacter baumannii*	3 (3.8)	6	0 (0.0)	0 (0.0)	3 (2.9)	6 (5.6)	9 (4.6)	0 (0.0)	9 (4.3)
*Enterobacter cloacae*	2 (2.6)	1	0 (0.0)	0 (0.0)	2 (2.0)	1 (0.9)	3 (1.5)	0 (0.0)	3 (1.4)
Others	8 (10.3)	0	0 (0.0)	0 (0.0)	8 (7.8)	0 (0.0)	8 (4.1)	0 (0.0)	8 (3.8)
*Candida* spp.	8 (10.3)	8	0 (0.0)	0 (0.0)	8 (7.8)	8 (7.6)	14 (7.1)	2 (16.7)	16 (7.7)
total	78 (100)	85 (100)	24 (100)	22 (100)	102 (100)	107	197 (100)	12 (100)	209 (100)

*n*, number of isolates; UTI, urinary tract infections; CA-UTI, catheter-associated UTIs; non-CA-UTI non-catheter-associated UTIs.

**Table 2 antibiotics-10-00007-t002:** Drug susceptibility of the most common etiological factors of UTIs in the Clinical Department of Orthopedic, Traumatic, Oncological, and Reconstructive Surgery in 2013–2015.

Antibiotics	*Escherichia coli* *n* = 61	*Klebsiella pneumoniae* *n* = 32	*Proteus mirabilis* (*n* = 25)	*Pseudomonas aeruginosa* *n* = 21
Beta-lactam antibacterials: penicillins, with extended-spectrum, beta-lactamase resistant penicillins, combinations of penicillins incl. beta-lactamase inhibitors				
Ampicillin	26%	0%	37%	NT
Piperacillin	28%	0%	35%	75%
amoxicillin + clavulanate	82%	35%	91%	NT
piperacillin + tazobactam	70%	31%	92%	85%
ticarcillin + clavulanate	NT	NT	NT	90
Other beta-lactam antibacterials: second/third-generation cephalosporins, carbapenems				
Cefuroxime	84%	25%	78%	NT
Ceftazidime	94%	38%	93%	85%
Cefotaxime	96%	38%	94%	NT
Cefepime	92%	31%	97%	100%
Imipenem	100%	100%	33%	77%
Meropenem	100%	100%	100%	77%
Ertapenem	100%	100%	100%	NT
Aminoglycoside antibacterials				
Gentamicin	87%	25%	67%	88%
Tobramycin	73%	36%	60%	91%
Amikacin	88%	50%	67%	92%
Netilmicin	89%	40%	60%	82%
Quinolone antibacterials				
Ciprofloxacin	64%	42%	57%	37%
Levofloxacin	64%	38%	43%	50%
Other antibacterials				
Nitrofurantoin	96%	NT	NT	NT
Fosfomycin	100%	80%	NT	98%
trimethoprim-sulfamethoxazole	55%	42%	50%	NT

*n*—total isolates, NT: not tested.

**Table 3 antibiotics-10-00007-t003:** Multidrug-resistant micro-organisms isolated from catheter-associated urinary tract infections in the Clinical Department of Orthopedic, Traumatic, Oncological, and Reconstructive Surgery in 2013–2015.

Pathogen	MDR Isolates [*n*]			Prevalence of MDR [%]	Trend in 2013–2015 *p*-Value
	Female	Male	Total		
*Acinetobacter baumannii* XDR, *n* = 9	2	5	7	77.8	*p* = 0.6015
*Klebsiella pneumoniae* MDR, incl. ESBL, *n* = 32	10	11	21	65.6	*p* = 0.6015
*Pseudomonas aeruginosa* MDR, XDR, *n* = 21	2	3	5	23.8	*p* = 0.6015
*Proteus mirabilis* MDR, incl. ESBL, *n* = 25	2	2	4	16.0	*p* = 0.2008
*Escherichia coli* MDR, incl. ESBL, *n* = 61	2	6	8	13.1	*p* = 0.6015
Others, *n* = 61	6	2	6	9.8	*p* = 0.6015
Total, *n* = 209	20	31	51	24.4	*p* = 0.6015

ESBL—extended-spectrum beta-lactamases; MDR—multidrug-resistant; XDR—extensively drug-resistant.

## Data Availability

The datasets generated or analyzed during this study are available and can be obtained, at request, from Iwona Pawłowska (e-mail: ivi5@op.pl) on reasonable enquiry.

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
