# Peer review of "Multidrug-Resistant Micro-Organisms Associated with Urinary Tract Infections in Orthopedic Patients: A Retrospective Laboratory-Based Study"

_antibiotics, 2020, doi:10.3390/antibiotics10010007_

Round 1

Reviewer 1 Report

Dear Authors

Thank you very much for your manuscript submission.

This investigation is a remarkable one. However, some confused methodological calculations are seen:

In Materials and Methods section: The authors should mention the months of the years, too.

In Results section: The authors mentioned 168 UTI cases (totally; in which 158 of them were CAUTI cases); then they mentioned 209 microbial strains were isolated. In this case, it seems that the recognized UTIs are poly-microbial infections.

The authors have represented the percentage of the infections and the related microorganisms in accordance with 209 strains. While it should be calculated in accordance with UTI and CAUTI cases. The CRITERION for calculation is 168 and not 209.

The authors should show the rate of poly-microbial UTIs in their investigation.

Due to this fact the tables should be revised and need precise explanations and legends.

Again, I emphasize that this work is invaluable, however some designed plans should be completely revised.

In this regard, the discussion and conclusion sections should be revised too.

Hence, my decision regarding this work is "Reconsider after Major Revision".

Best Regards

Author Response

Dear Sir/Madame

Thank you for your review. We have refered to all your suggestions, details below.

“Dear Authors

Thank you very much for your manuscript submission.

This investigation is a remarkable one. However, some confused methodological calculations are seen:

In Materials and Methods section: The authors should mention the months of the years, too. ( line 78)

In Results section: The authors mentioned 168 UTI cases (totally; in which 158 of them were CAUTI cases); then they mentioned 209 microbial strains were isolated. In this case, it seems that the recognized UTIs are poly-microbial infections.( line 128-9)

The authors have represented the percentage of the infections and the related microorganisms in accordance with 209 strains. While it should be calculated in accordance with UTI and CAUTI cases. The CRITERION for calculation is 168 and not 209. (line 133-4, 137, 336)

The authors should show the rate of poly-microbial UTIs in their investigation (line 133-4).

Due to this fact the tables should be revised and need precise explanations and legends.(line 336)

Again, I emphasize that this work is invaluable, however some designed plans should be completely revised.

In this regard, the discussion and conclusion sections should be revised too.

Hence, my decision regarding this work is "Reconsider after Major Revision".

Best Regards”

Sincerely

Reviewer 2 Report

This is a valuable study endeavor that aims to provide a thorough microbiological assessment of urinary tract infections in orthopedic surgery patients, with the special emphasis on resistance traits. However, there are some issues that have to be addressed before this manuscript can be considered for publication in the journal Antibiotics.

First and foremost, the whole manuscript should be additionally proofread by a native English speaker (for example, sentences should never begin with a number, like the first sentence in the Results section, and there is also a myriad of small errors). Also, once the microorganism name is abbreviated, only shortened version should be used further in the text. Finally, all abbreviations should be initially stated in full (e.g., this is not the case for NHSN).

I suggest modifying the manuscript title by removing a number of patients (this is something that is important to have in the Abstract, not the headline). In the Introduction (Background) section, study aims should be improved; more specifically, the authors state that the use of catheterization procedures was analyzed for patients both in the perioperative period and for catheterization for other reasons, but the manuscript does not address this appropriately.

In the 'Materials and Methods' section, country should also be added when describing the location of the hospital. Type of antibiotic susceptibility test should be stated (it is probably a microdilution method, but it is not stated explicitly). Also, there is no information whether disc diffusion method was also pursued, which is a possibility because double disk synergy test is mentioned. Likewise, double disk synergy test should be additionally explained how it exactly works.

The details of the used statistical package developer should be stated (akin to automated system used in microbiological part of the methodology). Furthermore, it is stated that p = 0.05 was considered significant, but more clarity should be provided, as also values lower than 0.05 are statistically significant, and it should be highlighted whether this is one-sided or two-sided.

One of the biggest weakness of this study is that there is not even basic sociodemographic data of patients included in this study. It would be valuable to know at least sex and mean/median/IQR age of patients, and if possible, to conduct a breakdown of isolates according to these categories. This would increase the informative value of this manuscript for further reference.

In the Discussion section, study strengths and (especially) weaknesses should be discussed. The reference on frequent use of trimethoprim/sulfamethoxazole in Poland is missing. Some newer studies on the topic should also be cited (e.g. Gossett et al. Urinary Tract Infection after Total Hip Arthroplasty: A Retrospective Cohort Study. J Surg Orthop Adv 2020;29:162-164).

Author Response

Dear Sir/Madame

Thank you for your review. We have refered to all your suggestions, details below.

“This is a valuable study endeavor that aims to provide a thorough microbiological assessment of urinary tract infections in orthopedic surgery patients, with the special emphasis on resistance traits. However, there are some issues that have to be addressed before this manuscript can be considered for publication in the journal Antibiotics.

First and foremost, the whole manuscript should be additionally proofread by a native English speaker (for example, sentences should never begin with a number, like the first sentence in the Results section, and there is also a myriad of small errors). Also, once the microorganism name is abbreviated, only shortened version should be used further in the text. Finally, all abbreviations should be initially stated in full (e.g., this is not the case for NHSN). (We improved it)

I suggest modifying the manuscript title by removing a number of patients (this is something that is important to have in the Abstract, not the headline). (We changed a title)

In the Introduction (Background) section, study aims should be improved; more specifically, the authors state that the use of catheterization procedures was analyzed for patients both in the perioperative period and for catheterization for other reasons, but the manuscript does not address this appropriately. (line 81-82)

In the 'Materials and Methods' section, country should also be added when describing the location of the hospital. (line 80)

Type of antibiotic susceptibility test should be stated (it is probably a microdilution method, but it is not stated explicitly). Also, there is no information whether disc diffusion method was also pursued, which is a possibility because double disk synergy test is mentioned. Likewise, double disk synergy test should be additionally explained how it exactly works. (line 98-101, 104-110)

The details of the used statistical package developer should be stated (akin to automated system used in microbiological part of the methodology). Furthermore, it is stated that p = 0.05 was considered significant, but more clarity should be provided, as also values lower than 0.05 are statistically significant, and it should be highlighted whether this is one-sided or two-sided. (line 119-120).

One of the biggest weakness of this study is that there is not even basic sociodemographic data of patients included in this study. It would be valuable to know at least sex and mean/median/IQR age of patients, and if possible, to conduct a breakdown of isolates according to these categories. This would increase the informative value of this manuscript for further reference.( line 125-129)

In the Discussion section, study strengths and (especially) weaknesses should be discussed (line 213-217).

The reference on frequent use of trimethoprim/sulfamethoxazole in Poland is missing.( line 202-203)

 Some newer studies on the topic should also be cited (e.g. Gossett et al. Urinary Tract Infection after Total Hip Arthroplasty: A Retrospective Cohort Study. J Surg Orthop Adv 2020;29:162-164).” (line 59-60)

Sincerely

Round 2

Reviewer 1 Report

Dear Authors

Thank you very much for your revision. Although the revision is done hurriedly, the manuscript is much better now. The reader can concentrate on the main cases which should be revised. The cases which should be revised are as below:

  1. The authors have forgotten to mention the details of the sampling in materials and methods Section.
  2. The abbreviation of "Healthcare-Associated Infections Surveillance Network (HAI-Net)" should be moved from line 88 to 87.
  3. In lines 119-120 the authors have mentioned "Statistical significance was assumed at p< 0.05.We used IBM SPSS statistics (v. 24; IBM Corp., Armonk, NY, USA) for all statistical analyses." but this claim is not seen in the manuscript.
  4. The authors represent 168 UTI cases. But in their calculation, the criterion is 163. If you want to take the criterion of 163, so you have to explain the exclusion of 5 cases. Otherwise, your criterion will be 168 and not 163.
  5. You have done a great investigation with great data. So, represent your data regarding microbial agents of UTIs and CAUTIs separately in the format of a table. Add, the gender of the patients with UTIs and CAUTIs, too. The statistical significance should be used in this regard, too.
  6. The microbial agents in poly-microbial and mono-microbial UTIs and CAUTIs should be represented in the format of a table. Also, add the gender of the patients. The statistical significance should be used in this regard, too.
  7. The frequency of sensitive and resistant microbial agents and ESBL+/- strains should be represented in UTIs and CAUTIs and poly-microbial and mono-microbial cases. The statistical significance should be used in this regard, too. 
  8. So, the results section should be revised in accordance with aforementioned cases.
  9. These cases should be compared and discussed in Discussion Section.
  10. The conclusion should be revised.
  11. In the case of UTIs investigations, Gender, Microbial agents and the antibiotic sensitivity should be noticed as parallel and complementary items.
  12. Due to this fact my decision regarding your work is "Reconsider after major revision"

Best Regards

Author Response

DETAILED RESPONSE TO REVIEWERS:

STEP-BY-STEP REPLIES TOREVIEWERS' COMMENTS:

Reviewer #2:

Thank you very much for your revision. Although the revision is done hurriedly, the manuscript is much better now.

Authors’ reply: Thank you for this comment!

The reader can concentrate on the main cases which should be revised. The cases which should be revised are as below:

The authors have forgotten to mention the details of the sampling in materials and methods Section.

Authors’ reply: Corrected according to suggestions. The “MATERIALS AND METHODS” section was supplemented, as below (lines 97-98, all changes in the revised manuscript have been highlighted in yellow):

“(...) For the microbiological diagnosing of CA-UTI, urine samples were collected through newly placed catheters using aseptic technique. In the case of patients without catheters, mid-stream urine was collected. (...)”.

The abbreviation of "Healthcare-Associated Infections Surveillance Network (HAI-Net)" should be moved from line 88 to 87.

Authors’ reply: Corrected according to suggestions.

In lines 119-120 the authors have mentioned "Statistical significance was assumed at p< 0.05.We used IBM SPSS statistics (v. 24; IBM Corp., Armonk, NY, USA) for all statistical analyses." but this claim is not seen in the manuscript.

Authors’ reply: Corrected according to suggestions. The “RESULTS” section was supplemented, (lines: 132-133, 145-147, 151, 158, all changes in the revised manuscript have been highlighted in yellow).

The authors represent 168 UTI cases. But in their calculation, the criterion is 163. If you want to take the criterion of 163, so you have to explain the exclusion of 5 cases. Otherwise, your criterion will be 168 and not 163.

Authors’ reply: Corrected according to suggestions. The “RESULTS” section and table 1 were corrected (all changes in the revised manuscript have been highlighted in yellow).

You have done a great investigation with great data. So, represent your data regarding microbial agents of UTIs and CAUTIs separately in the format of a table. Add, the gender of the patients with UTIs and CAUTIs, too. The statistical significance should be used in this regard, too.

Authors’ reply: Corrected according to suggestions. The “RESULTS” section and table 1 and table 3 were supplemented and corrected (lines 130, 132-135, 149-151, table I, table III, all changes in the revised manuscript have been highlighted in yellow).

The microbial agents in poly-microbial and mono-microbial UTIs and CAUTIs should be represented in the format of a table. Also, add the gender of the patients. The statistical significance should be used in this regard, too.

Authors’ reply: Corrected according to suggestions. The “RESULTS” section and table 1 were supplemented and corrected (all changes in the revised manuscript have been highlighted in yellow). The limitation for the performance of static analysis was small number of non-CA-UTI and monomicrobial UTI.

The frequency of sensitive and resistant microbial agents and ESBL+/- strains should be represented in UTIs and CAUTIs and poly-microbial and mono-microbial cases. The statistical significance should be used in this regard, too. 

 Authors’ reply: Corrected according to suggestions. The “RESULTS” section and table 3 were supplemented and corrected (all changes in the revised manuscript have been highlighted in yellow). The limitation for the performance of static analysis was small number of non-CA-UTI and monomicrobial UTI.

So, the results section should be revised in accordance with aforementioned cases.

Authors’ reply: Corrected according to suggestions. Corrected according to suggestions. The “RESULTS” section and table 3 were supplemented and corrected (lines 127-162, all changes in the revised manuscript have been highlighted in yellow)

These cases should be compared and discussed in Discussion Section. The conclusion should be revised.

Authors’ reply: Corrected according to suggestions. Corrected according to suggestions. The “DISCUSSION” section supplemented (all changes in the revised manuscript have been highlighted in yellow)

In the case of UTIs investigations, Gender, Microbial agents and the antibiotic sensitivity should be noticed as parallel and complementary items. Due to this fact my decision regarding your work is "Reconsider after major revision"

Authors’ reply: Above all, thank you for great help in preparing the text!

Round 3

Reviewer 1 Report

Dear Authors

Thank you very much for your deep and precise revision. My decision regarding your work is "Accept in present form".

Best Regards